# Role of Base Strength, Cluster Structure and Charge in Sulfuric Acid-Driven Particle Formation

Nanna Myllys[1,2], Jakub Kubečka[2], Vitus Besel[2], Dina Alfaouri[2], Tinja Olenius[3], James N Smith[1], and Monica Passananti[2,4]

[1]Department of Chemistry, University of California, Irvine
[2]Institute for Atmospheric and Earth System Research, University of Helsinki
[3]Department of Environmental Science and Analytical Chemistry & Bolin Centre for Climate Research, Stockholm University
[4]Dipartimento di Chimica, Universitá di Torino

**Correspondence:** Nanna Myllys (nanna.myllys@uci.edu)

**Abstract.** In atmospheric sulfuric acid-driven particle formation, bases are able to stabilize the initial molecular clusters, and thus enhance particle formation. The enhancing potential of a stabilizing base is affected by different factors, such as the basicity and abundance. Here we use weak (ammonia), medium strong (dimethylamine) and very strong (guanidine) bases as representative atmospheric base compounds, and systematically investigate their ability to stabilize sulfuric acid clusters. Using quantum chemistry, we study proton transfer as well as intermolecular interactions and symmetry in clusters, of which the former is directly related to the base strength and the latter to the structural effects. Based on the theoretical cluster stabilities and cluster population kinetics modeling, we provide molecular-level mechanisms of cluster growth and show that in electrically neutral particle formation, guanidine can dominate formation events even at relatively low concentrations. However, when ions are involved, charge effects can stabilize small clusters also for weaker bases. In this case the atmospheric abundance of the bases becomes more important, and thus ammonia is likely to play a key role. The theoretical findings are validated by cluster distribution experiments, as well as comparisons to previously reported particle formation rates, showing a good agreement.

## 1 Introduction

Atmospheric aerosol particles influence human health and global climate (Kulmala et al., 2007). Airborne particles act as condensation nuclei for clouds and can also directly absorb or scatter the incoming radiation, forming a significant but highly uncertain effect on Earth's radiation balance. New-particle formation (NPF) from atmospheric vapors is a significant source of ultrafine particles, but all the participating vapors as well as the molecular-level mechanisms are not fully resolved (Zhang et al., 2012; Hallquist et al., 2009). In the present-day atmosphere that contains high levels of sulfur, sulfuric acid is a key precursor vapor and has been shown to be linked to new-particle formation events in various environments. However, sulfuric acid-driven NPF requires additional stabilizing compounds in order to yield particle formation rates similar to those observed in the atmosphere (Kulmala et al., 2013). These compounds include atmospheric bases and ions (Almeida et al., 2013; Lehtipalo et al., 2016).

The most abundant base in the atmosphere is ammonia with a typical gas-phase concentration at the level from sub-ppb$_V$ to tens of ppb$_V$. A major source of ammonia is agricultural emissions, with other important sources including industry, oceans and vegetation (Anderson et al., 2003). Ammonia has been shown to significantly increase particle formation rates in comparison to the binary sulfuric acid–water system, and is thus expected to be an important player in NPF in at least some environments (Kurtén et al., 2007). Ammonia is a weak base with a dissociation constant p$K_b$ of 4.75 and a gas-phase basicity of $-195.7$ kcal/mol, and can stabilize sulfuric-acid-containing molecular clusters by proton transfer reactions and hydrogen bond formation. Amines, on the other hand, are stronger bases than ammonia, and show a much larger stabilization effect (Almeida et al., 2013). Approximately 150 amines have been detected in the atmosphere, with alkylamines being the most abundant at the level of ppt$_V$ (Ho et al., 2008). Amine emissions are dominated by human activities such as industry, animal husbandry and fish processing, with common natural sources being soils and marine environments (Ge et al., 2011). In recent years, dimethylamine has been the most studied amine in atmospheric particle formation research. It is a medium-strong base with a p$K_b$ value of 3.27 and a gas-phase basicity of $-214.3$ kcal/mol. Dimethylamine has been found to enhance new-particle formation in various environments, including Hyytiälä boreal forest in Finland and Shanghai megacity in China (Kulmala et al., 2013; Yao et al., 2018). Also laboratory experiments and computational studies have confirmed that dimethylamine is able to enhance sulfuric acid-driven particle formation rates by up to several orders of magnitude compared to ammonia (Almeida et al., 2013; Olenius et al., 2013; Kurtén et al., 2008; Ahlm et al., 2016; Temelso et al., 2018).

In addition of the commonly studied ammonia and amines, several studies have recently investigated possibilities of other bases to participate in new-particle formation. For instance, diamines, amineoxides and guanidine compounds have been suggested to have a role in the stabilization of sulfuric-acid-containing clusters (Xie et al., 2017; Jen et al., 2016; Elm et al., 2016; Myllys, 2017). In fact, these compounds are able to enhance particle formation much more effectively than ammonia or dimethylamine, however, their atmospheric abundances remain unclear. Multifunctional compounds such as diamines and amineoxides can form more intermolecular interactions than monoamines, and thus the heterodimer formation from acid and base molecules as well as the subsequent cluster growth are more efficient (Elm et al., 2017). Extremely strong organobases, such as guanidine compounds, may interact with sulfuric acid so strongly that the evaporation of clusters is negligible. In this case particle formation becomes fully collision-driven, i.e. occurs without thermodynamic barriers. In our recent computational study, we demonstrated that at similar ambient conditions, guanidine can enhance $\sim$1-nm nanoparticle formation rates by up to several orders of magnitude compared to dimethylamine. We also showed that guanidine requires a significantly lower gas-phase base concentration ($\sim$2000 times lower) to reach the same enhancing effect on molecular cluster formation as dimethylamine (Myllys et al., 2018). This implies that even at a very low atmospheric concentration, strong bases might have an important role in the initial steps of particle formation.

There exists a plethora of strong base species, and here we use guanidine, with a p$K_b$ value of 0.4 (Angyal and Warburton, 1951) and a gas-phase basicity of $-226.9$ kcal/mol, as a representative strong base. Guanidine may be released to the environment through various waste streams, including the production and use in industry in the manufacture of, e.g., medicines, military munitions, polymeric resins and flame retardants (Kumar et al., 2002; Oxley et al., 2008; Zhao et al., 2015; Kaplan et al., 1982). In addition, guanidine can be released from natural sources as it is a normal product of protein metabolism

(Marescau et al., 1992; Bonas et al., 1963; Van Pilsum et al., 1956; Swick, 1958). As guanidine is a strong base, its volatilization from wet environments can be assumed to be negligible due to guanidinium cation formation. However, the saturation vapor pressure of neutral guanidine is 293 Pa (at room temperature) which indicates that it is likely to volatilize from dry surfaces (The Merck Index, 2013).

Ions, a focus of the current study, can enhance cluster binding through strong intermolecular bond formation with electrically neutral molecules. The bisulfate anion or the protonated base in charged sulfuric acid–base clusters can act as a strong conjugate base or acid and suppress the evaporation of especially the smallest clusters. Ions can thus play an important role in the initial steps of NPF, but their relative enhancement with respect to cluster formation from solely electrically neutral molecules depends on the stability of the neutral clusters (Lehtipalo et al., 2016). In addition, charged species can be directly detected by mass

spectrometer techniques, which enables direct comparison of measurements and molecular modeling.

In this paper we apply computational chemistry to comprehensively and systematically investigate the effect of base properties on two-component sulfuric acid–base nanoparticle formation. We consider the strength and abundance of the base, and use ammonia, dimethylamine and guanidine as proxies for weak, medium strong and very strong bases, respectively. We study the role of ion-mediated particle formation in the different sulfuric acid–base systems by including negatively and positively

charged clusters containing a bisulfate anion or a base cation. Electrospray Ionization Atmospheric Pressure interface Time-Of-Flight (ESI-APi-TOF) measurements are performed to further confirm the theoretical findings.

## 2    Computational and Experimental Details

### 2.1    Gibbs Free Energy of Cluster Formation

Determining atmospheric cluster stabilities and their effects on cluster formation kinetics requires calculating the Gibbs free

formation energies. It is generally assumed that the global minimum free energy structures of different cluster compositions dominate atmospheric cluster distributions, and can thus be used to describe the properties of a cluster population. For clusters consisting of several molecules, the potential energy surface becomes highly complicated and finding the global minimum free energy structure is challenging. Here we study acid–base clusters containing 0–4 acid and 0–4 base molecules, including both electrically neutral clusters as well as the corresponding anionic and cationic clusters. We used cluster structures of our

previous studies (Myllys et al., 2018, 2019; Olenius et al., 2013) as a basis for global minimum Gibbs free energy clusters. The structures of clusters not studied before were obtained by a new configurational sampling procedure, as explained in the supporting information. For previously reported cluster structures that seemed to differ from general trends, we conducted a new configurational sampling to test if the global minimum had been found correctly. For anionic clusters, we include compositions in which the number of acid molecules is equal or larger than the number of base molecules, and for cationic

clusters compositions that have an equal or larger number of base molecules compared to acid molecules. This selection saves computational time without affecting the particle formation modeling results, as other types of compositions can be expected to be less stable and thus redundant.

Cluster geometries are optimized and the vibrational frequencies are calculated using $\omega$B97X-D/6-31++G** level of theory (Chai and Head-Gordon, 2008; Krishnan et al., 1980). In previous studies, $\omega$B97X-D/6-31++G** has been shown to yield good geometries and thermochemical parameters for non-covalently bound molecular clusters (Myllys et al., 2016b). In order to obtain highly accurate binding energies, we calculate electronic energy corrections on top of the DFT structures using a linear-scaling coupled cluster method DLPNO–CCSD(T) with an aug-cc-pVTZ basis set (Riplinger and Neese, 2013; Riplinger et al., 2013, 2016; Kendall et al., 1992). We use tight pair natural orbital criteria, tight self consistent field criteria and integration grid 4 in all coupled cluster calculations (keywords TightPNO, TightSCF, GRID4) (Liakos et al., 2015). We have shown earlier that the DLPNO–CCSD(T)/aug-cc-pVTZ level of theory with TightPNO yields binding energies close to the canonical coupled clusters with a significant gain in computational resources (Myllys et al., 2016a, 2018). All geometries are optimized and vibrational frequencies are calculated using Gaussian 16 RevA.03 (Frisch et al., 2016). Electronic energy corrections are performed in Orca version 4.0.1.2. (Neese, 2012). Thermochemistry is calculated using rigid rotor–harmonic oscillator approximation and Gibbs free energies are presented in kcal/mol and at 298.15 K. For simplicity, we refer to sulfuric acid as A, ammonia as N, dimethylamine as D and guanidine as G, and cluster compositions as e.g. 2D3A, which refers to a cluster of two dimethylamine and three sulfuric acid molecules.

## 2.2 Atmospheric Cluster Dynamics Code

To study cluster formation kinetics and the dynamics of cluster populations, the calculated Gibbs free energies are used as input in Atmospheric Cluster Dynamics Code (ACDC) (McGrath et al., 2012). The detailed theory of ACDC is explained in the supporting information. Briefly, the model simulates nanoparticle formation by solving the cluster distribution considering collision, evaporation and removal processes. The model calculates the rate constants for each process among the population of clusters and vapor molecules, and solves the cluster birth–death equations at given conditions.

## 2.3 ESI-APi-TOF MS Measurements

Charged sulfuric acid clusters with ammonia, dimethylamine and guanidine were generated in laboratory experiments using an electrospray ionizer (ESI) and analysed by an Atmospheric Pressure interface Time Of Flight Mass Spectrometer (APi-TOF MS). Three samples were prepared and used to generate charged clusters: 100 mmol/l sulfuric acid with 100 mmol/l dimethylamine, 100 mmol/l sulfuric acid with 100 mmol/l guanidine, and 100 mmol/l ammonium bisulfate, all the solutions were prepared in 50% methanol and 50% of Milli-Q water. The solutions were sprayed in both negative and positive modes, producing negatively and positively charged clusters, respectively. The charged clusters were detected by the APi-TOF (Tofwerk AG) mass spectrometer operating in both polarities accordingly. The data were analysed using the Matlab-based program TofTools developed at the University of Helsinki. Further details about the APi-TOF and TofTools can be found from study of Junninen et al. (2010).

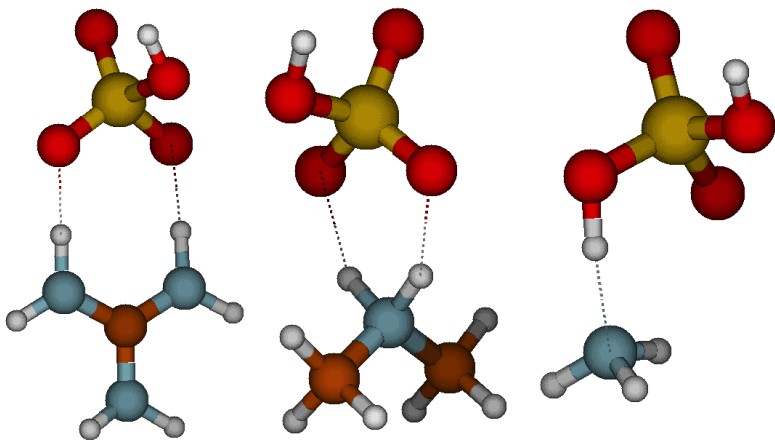

**Figure 1.** Molecular structures of sulfuric acid heterodimers with guanidine (left), dimethylamine (middle) and ammonia (right). Color coding: brown is carbon, blue is nitrogen, red is oxygen, yellow is sulfur and white is hydrogen.

## 3 Results and Discussion

### 3.1 Acid–Base Heterodimer Formation

The formation of an acid–base heterodimer has been shown to be a crucial step in initial particle formation for many molecular systems (Elm, 2017). Figure 1 shows the molecular structures of the studied heterodimers. In the case of the guanidine and
5   dimethylamine complexes, the proton has transferred from sulfuric acid to base and there are two intermolecular interactions between the acid and the base. Whereas in the guanidine–sulfuric acid complex the hydrogen bonds are linear, i.e. the donor-hydrogen-acceptor angles are close to $180°$, in the dimethylamine–sulfuric acid complex the bond angles are $145$–$150°$ which decreases the intermolecular interaction strength compared to straight angles. Therefore, the intermolecular bonds between guanidine and sulfuric acid are much stronger than those between dimethylamine and sulfuric acid. Since ammonia is a weak
10   base, there is no proton transfer in the heterodimer. The ammonia and sulfuric acid molecules form a complex via one hydrogen bond, leading to the binding of sulfuric acid to ammonia being weaker than to dimethylamine or guanidine.

     The molecular interaction between the acid and base molecules defines the stability of a formed heterodimer and accordingly its theoretical maximum concentration at given conditions assuming an equilibrium situation. Assuming mass-balance relation for the heterodimer formation reaction leads to the following concentration under equilibrium conditions:

$$15 \quad [(\mathrm{acid})(\mathrm{base})] = [\mathrm{acid}][\mathrm{base}]\frac{k_{\mathrm{B}}T}{P_{\mathrm{ref}}}\exp\left(-\frac{\Delta G_{\mathrm{ref}}}{k_{\mathrm{B}}T}\right) \tag{1}$$

The equilibrium concentration $[(\mathrm{acid})(\mathrm{base})]$ of the heterodimer is dependent both on the Gibbs free formation energy $\Delta G_{\mathrm{ref}}$ (calculated at reference pressure $P_{\mathrm{ref}}$) at given temperature $T$ and on the monomer concentrations $[\mathrm{acid}]$ and $[\mathrm{base}]$. Now we can study how large the magnitude of the exponential Gibbs free energy contribution has relative to the linear concentration factors. The Gibbs free formation energies (at 298.15 K) are $-6.8$ kcal/mol for ammonia–sulfuric acid, $-13.5$ kcal/mol for

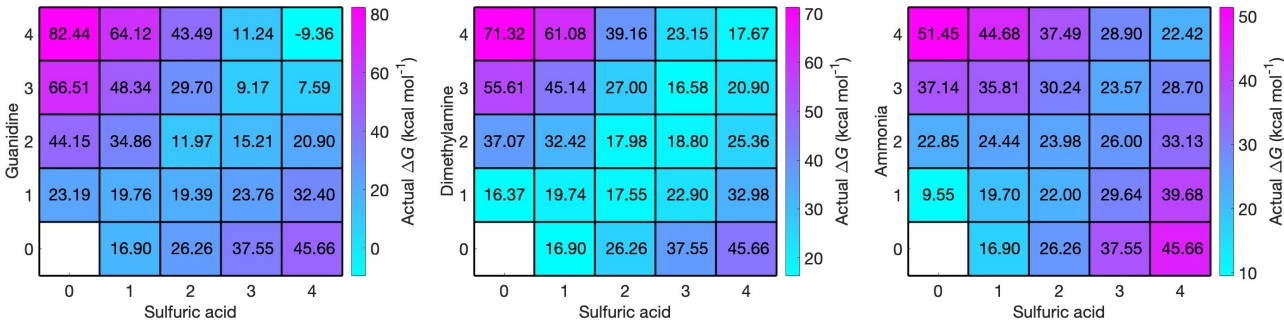

**Figure 2.** Vapor concentration-dependent Gibbs free energies for electrically neutral acid–base clusters at 298.15 K. Sulfuric acid concentration is $10^7$ cm$^{-3}$ in all cases, and for bases the relative concentrations of [guanidine]$=10^{-5}$ ppt$_V$ (left), [dimethylamine]$=1$ ppt$_V$ (middle) and [ammonia]$=10^5$ ppt$_V$ (right) are used.

dimethylamine–sulfuric acid, and $-20.3$ kcal/mol for guanidine–sulfuric acid dimers. Assuming the same sulfuric acid concentration in all cases, we can calculate what the relative concentrations of ammonia, dimethylamine and guanidine should be to yield the same heterodimer concentration, and obtain [G] $\approx 1$, [D] $\approx 10^5$ and [N] $\approx 10^{10}$. This means that if the atmospheric ammonia concentration is $10^5$ ppt$_V$, 1 ppt$_V$ of dimethylamine or $10^{-5}$ ppt$_V$ of guanidine is required to yield a same heterodimer equilibrium concentration as in the case of ammonia. We will refer to these concentrations as relative base concentrations throughout the text.

We have calculated the actual vapor concentration-dependent Gibbs free energies, obtained from the reference values $\Delta G_{\text{ref}}$ and vapor concentrations through the law of mass action (Eq. (1)), for all acid–base cluster compositions at the relative base concentrations and at a sulfuric acid monomer concentration of $10^7$ cm$^{-3}$. At these concentrations, the vapor-dependent Gibbs free energy for all acid–base heterodimers is the same, but Figure 2 shows that further cluster growth is most favorable for guanidine even if its concentration is 5 and 10 orders of magnitude lower than that of dimethylamine and ammonia, respectively. These results demonstrate that, in terms of thermodynamics, the enhancement potential of base in sulfuric-acid-driven clustering is largely dominated by the base strength (characterized by $\Delta G_{\text{ref}}$), and the relative concentration plays only a minor role.

The thermodynamically most favorable clustering pathway for all acid–base systems is close to the diagonal axis, i.e. the actual Gibbs free energy exhibits its lowest values when the number of acid and base molecules is equal, or when the difference between the numbers of acid and base molecules is one. The heterodimer evaporation rates are $10^5$ s$^{-1}$ for 1N1A, 1 s$^{-1}$ for 1D1A and $10^{-5}$ s$^{-1}$ for 1G1A (see the supporting information). This implies that the lifetime of 1N1A is very short and even at an ammonia concentration as high as 100 ppb$_V$, it is unlikely that the concentration of 1N1A heterodimers would be high enough for these clusters to contribute to further cluster growth by coagulation processes. Instead, the growth can be expected to occur via monomeric acid and base additions. The 1D1A cluster has five orders of magnitude lower evaporation rate compared to 1N1A, and this heterodimer is relatively stable. However, because the equilibrium concentration is more than

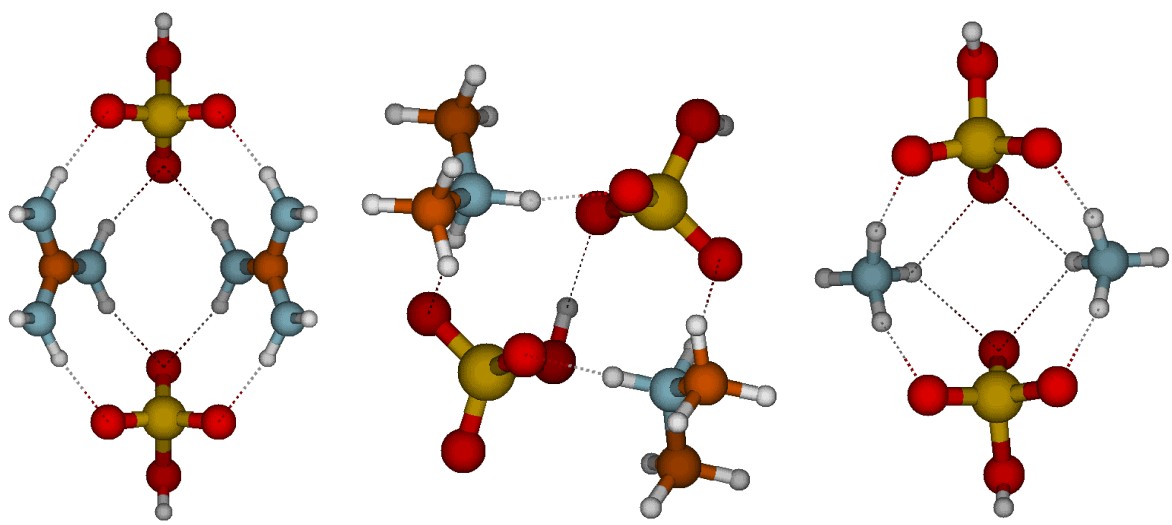

**Figure 3.** Molecular structures of clusters containing two sulfuric acid and two base molecules for guanidine (left), dimethylamine (middle) and ammonia (right). Color coding: brown is carbon, blue is nitrogen, red is oxygen, yellow is sulfur and white is hydrogen.

two orders of magnitude lower than that of monomers, cluster collisions with monomers are still much more likely than those involving 1D1A clusters. The evaporation rate of the 1G1A heterodimer is very low, and therefore heterodimer coagulations are expected to make a major contribution to the growth of sulfuric acid–guanidine clusters. Since each addition of 1G1A to a pre-existing diagonal cluster leads to a lower actual free energy and the cluster evaporation is negligible, the only limiting
factor to particle formation in this system is the collision frequency between sulfuric acid and guanidine molecules.

### 3.2   Diagonal Cluster Structures

The reason that the clusters along the diagonal are most stable is shown in the cluster structures (Figure 2), in which all sulfuric acid molecules are able to donate a proton to a base molecule. The intermolecular interactions between bisulfate anions and protonated base cations are stronger than those between molecules with no proton transfers. Figure 3 shows the molecular
structures of 2(acid)2(base) clusters. In all cluster structures, there are two proton transfer reactions from sulfuric acid to base. Ammonia- and guanidine-containing clusters resemble each other in the way that there are eight intermolecular interactions between bisulfate and guanidinium or ammonium ions, the hydroxyl groups of both bisulfates remain free, and the structures have a $C_{2v}$ symmetry. In the 2G2A structure, the hydrogen bond angles are 160° in the inner circle and 170° in the outer circle, whereas in the 2N2A cluster they are 120 and 160°, respectively. This means that in the 2N2A cluster, the hydrogen bonds in
the inner circle are very weak. The molecular structure of the 2D2A cluster differs remarkably from that of 2G2A and 2N2A: 2D2A contains five intermolecular interactions and one of them is between the two bisulfates through the free hydroxyl group and the oxygen atom moieties.

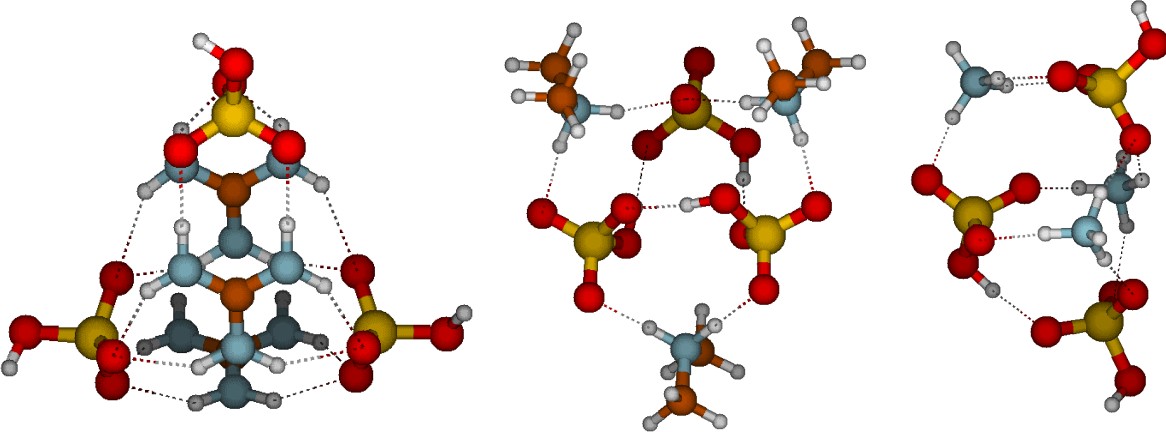

**Figure 4.** Molecular structures of clusters containing three sulfuric acid and three base molecules for guanidine (left), dimethylamine (middle) and ammonia (right). Color coding: brown is carbon, blue is nitrogen, red is oxygen, yellow is sulfur and white is hydrogen.

The Gibbs free binding energies are $-28.9$ kcal/mol for 2N2A, $-48.6$ kcal/mol for 2D2A and $-68.2$ kcal/mol for 2G2A. For 2N2A and 2D2A the dominant evaporation channel is the decomposition to 2(acid)1(base) + free base parties, with evaporation rate constants of $5 \times 10^4$ and $3 \times 10^{-3}$ s$^{-1}$, respectively. For the 2G2A cluster, the main decomposition pathway is different: the evaporation of a base molecule would require a proton transfer and breaking of four strong intermolecular interactions, whereas breaking into two 1G1A parts does not require proton transfer reactions but only the breaking of four intermolecular interactions. The dominant evaporation pathway for 2G2A is thus decomposition into heterodimers, with a rate constant of $3 \times 10^{-11}$ s$^{-1}$.

All 3(acid)3(base) clusters exhibit three proton transfers (Figure 4). In the 3N3A cluster structure, each ammonium ion forms three intermolecular interactions with a bisulfate. In addition there is one intermolecular bond between bisulfate anions. The main decomposition pathway, with a rate constant of $30$ s$^{-1}$, is via ammonia evaporation which requires one proton transfer and the breaking of three intermolecular interactions. In the 3D3A structure, each dimethylaminium interacts with two bisulfates via two intermolecular bonds. In addition, all bisulfates interact with two other bisulfates and thus each bisulfate forms four intermolecular bonds. The main evaporation route of 3D3A is via evaporation of dimethylamine at a rate of $4 \times 10^{-4}$ s$^{-1}$, requiring that a dimethylaminium donates a proton back to bisulfate and two intermolecular interactions are broken. In the case of guanidine-containing clusters, two guanidinium and two bisulfate ions form six intermolecular bonds and one guanidinium and one bisulfate form only four. Assuming that hydroxyl groups can freely rotate at room temperature, the 3G3A cluster is C$_\text{s}$ symmetric. The main evaporation pathway for 3G3A is the decomposition into 1G1A and 2G2A, which requires breaking six intermolecular bonds, and the evaporation rate is $3 \times 10^{-7}$ s$^{-1}$.

Finally, Figure 5 presents the molecular structures of 4(acid)4(base) clusters, in which four proton transfer reactions occur. In the case of the 4N4A cluster, all ammonium ions form three intermolecular bonds with bisulfate and *vice versa*. In the 4D4A cluster each bisulfate anion interacts with another bisulfate via two intermolecular bonds and the cluster contains a centre

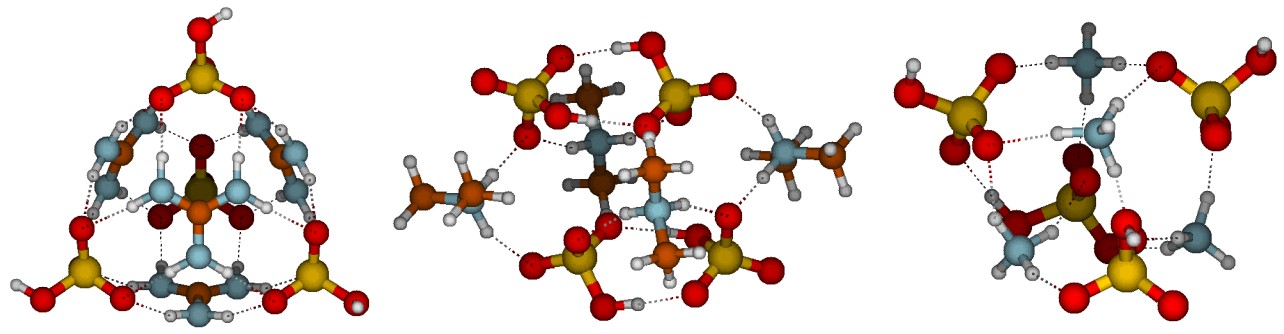

**Figure 5.** Molecular structures of clusters containing four sulfuric acid and four base molecules for guanidine (left), dimethylamine (middle) and ammonia (right). Color coding: brown is carbon, blue is nitrogen, red is oxygen, yellow is sulfur and white is hydrogen.

of inversion, thus belonging to the $C_i$ point group. All dimethylaminium ions form two intermolecular bonds with bisulfate moieties. In the 4G4A structure, each bisulfate interacts with three guanidinium ions and forms a total of six intermolecular bonds, and *vice versa*. The structure is mesh-like with free hydroxyl groups pointing out of the core. Assuming free rotation of hydroxyl groups at room temperature, the 4G4A cluster belongs to the $T_d$ point group. The main decomposition pathway of

5 the 4N4A cluster is the evaporation of ammonia with a rate of $6 \times 10^{-2}$ s$^{-1}$. 4D4A has two equally fast decomposition routes, evaporation into 1D1A + 3D3A or into two 2D2A parts, and its total evaporation rate is $7 \times 10^{-4}$ s$^{-1}$. The main evaporation pathway for 4G4A is to decompose into two 2G2A parts at a rate of $2 \times 10^{-15}$ s$^{-1}$. The overall evaporation rates for all clusters are presented in the supporting information in Figure S1.

We have simulated electrically neutral particle formation rates based on the calculated Gibbs free energies using the ACDC

model and compared the results to atmospheric measurements (see the supporting information Figure S2). We investigated which simulated base concentrations yield NPF rates close to the atmospheric observations when including only electrically neutral two-component clusters. We found that guanidine concentrations of 0.001–1 ppt$_V$, dimethylamine concentrations of 0.1–100 ppt$_V$ and ammonia concentrations of $10^4$–$10^7$ ppt$_V$ are needed to yield NPF rates of the magnitude of the observations. However, these results do not take ions or hydration into account, which are expected to increase the particle formation rate,

especially in the case of ammonia. In addition, synergistic effects between different bases may play a role in the atmosphere. For example, it has been demonstrated that the presence of ammonia increases particle formation when added to a two-component sulfuric acid–amine system (Myllys et al., 2019; Yu et al., 2012; Glasoe et al., 2015).

### 3.3 The Role of Ions in the First Steps of Acid–Base Particle Formation

In addition to proceeding through electrically neutral pathways, atmospheric cluster formation can be ion-induced (Wagner

et al., 2017; Kirkby et al., 2016). Sulfuric acid can be deprotonated in the atmosphere by generic air ions to form a bisulfate, here referred to as B. A bisulfate can form a 1A1B complex with a neutral sulfuric acid molecule, the formation free energy of which is highly exergonic (−33.8 kcal/mol), corresponding to an evaporation rate as low as $10^{-14}$ s$^{-1}$. Thus a large fraction of

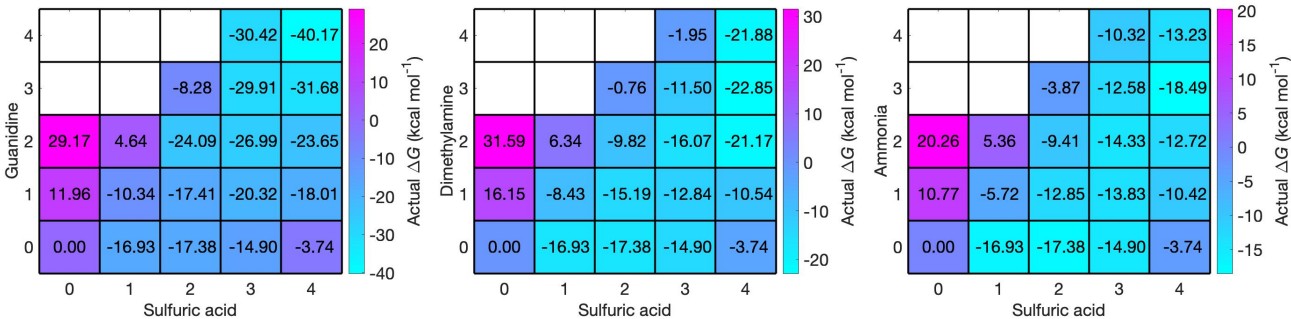

**Figure 6.** Concentration-dependent Gibbs free energies for bisulfate-containing acid–base clusters at 298.15 K. Sulfuric acid concentration is $10^7$ cm$^{-3}$ in all cases, and for bases the relative concentrations of [guanidine]=$10^{-5}$ ppt$_V$ (left), [dimethylamine]=1 ppt$_V$ (middle) and [ammonia]=$10^5$ ppt$_V$ (right) are used. Note that the $x$ and $y$ axes refer to the numbers of neutral acid and base molecules and each cluster contains one bisulfate anion.

bisulfate can be expected to exist as a complex with sulfuric acid, and this complex can grow further by uptake of acid or base molecules. The addition of a second sulfuric acid molecule is the most favorable reaction with a formation free energy of $-17.4$ kcal/mol. The reaction free energies for addition of guanidine, dimethylamine or ammonia are $-16.6$ kcal/mol, $-7.9$ kcal/mol, and 1.7 kcal/mol, respectively. This means that the only reaction competitive to the addition of sulfuric acid is the addition of

5 guanidine. The total evaporation rates of the resulting complexes are $10^{-2}$ s$^{-1}$ for 2A1B, $5 \times 10^{-2}$ s$^{-1}$ for 1A1B1G, $8 \times 10^4$ s$^{-1}$ for 1A1B1D and $10^{12}$ s$^{-1}$ for 1A1B1N. We can again study the vapor-dependent Gibbs free energies using the relative base concentrations, shown in Figure 6. The thermodynamically most favorable formation pathways of negative sulfuric acid–base clusters are below the diagonal axis, corresponding to clusters that contain more acid than base molecules. In the case of guanidine, there is no thermodynamic barrier for cluster growth along the most favorable pathway, and for dimethylamine

and ammonia only small barriers can be found around compositions including 1 bisulfate ion, 1 base and 2–3 sulfuric acid molecules. This indicates that the enhancing effect of bisulfate in particle formation becomes more remarkable for weaker bases, since the presence of bisulfate removes the thermodynamic barrier of cluster growth (which does not exist in the case of guanidine).

The base molecules can become ionized in the atmosphere by receiving a proton, here referred to as P, and form guanidinium

1G1P, dimethylaminium 1D1P and ammonium 1N1P cations. Protonated bases are likely to form a complex with their own conjugate base, the formation free energies for which are $-18.3$ kcal/mol for 2G1P, $-15.7$ kcal/mol for 2D1P and $-19.1$ kcal/mol for 2N1P. These protonated base dimers are likely to uptake a sulfuric acid molecule with reaction free energies of $-31.0$ kcal/mol for 1A2G1P, $-24.9$ kcal/mol for 1A2D1P and $-16.5$ kcal/mol for 1A2N1P, and the evaporation rates for these clusters are $10^{-10}$ s$^{-1}$, $4 \times 10^{-8}$ s$^{-1}$ and $7 \times 10^{-2}$ s$^{-1}$, respectively. All $n$(acid)$n$(base)1(protonated base) clusters are stable

against evaporation for guanidine and dimethylamine. Positive acid–ammonia clusters are somewhat less stable, but still have considerably lower evaporation rates than their neutral equivalents. Figure 7 shows the concentration-dependent Gibbs free

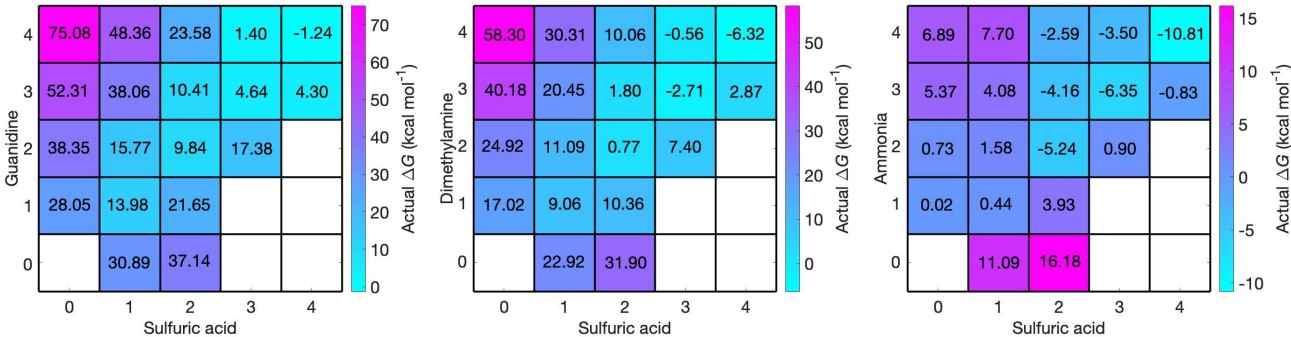

**Figure 7.** Concentration-dependent Gibbs free energies for protonated-base-containing acid–base clusters at 298.15 K. Sulfuric acid concentration is $10^7$ cm$^{-3}$ in all cases, and for bases the relative concentrations of [guanidine]=$10^{-5}$ ppt$_V$ (left), [dimethylamine]=1 ppt$_V$ (middle) and [ammonia]=$10^5$ ppt$_V$ (right) are used. Note that the $x$ and $y$ axes refer to the numbers of neutral acid and base molecules and each cluster contains one protonated base cation.

energies of these positively charged clusters at the relative base concentrations. In all cases, the lowest free energy path is on the diagonal axis, and the cluster formation along it is barrierless, although some barriers are related to growth around the diagonal compositions by monomer additions. The enhancing effect of ions in particle formation becomes more remarkable for weaker bases, since the presence of ions allows the cluster growth to occur without thermodynamic barrier (which does not exist in the case of guanidine). The enhancing effect of ions on particle formation rates is presented in Figure 10.

### 3.4 Charged Cluster Distributions

In experiments performed at the Cosmics Leaving OUtdoor Droplets (CLOUD) chamber as well as in other chamber experiments the clusters involved in NPF are often detected using an APi-TOF or a Chemical Ionization APi-TOF mass spectrometer (Almeida et al., 2013). However, only charged clusters can be directly detected by MS, therefore understanding the stability of both charged and neutral clusters can help to interpret the experimental data. Figure 8 shows the relative ionic acid–base cluster abundance measured using ESI-APi-TOF in the laboratory studies of this work. Since the charged clusters are produced from the liquid phase (Section 2.3), the absolute gas-phase concentrations are challenging to define accurately. Therefore, we use the cluster ratio instead of the absolute concentration to characterize the cluster distribution. The cluster ratio for negatively charged clusters is calculated as $\frac{[\text{cluster}]}{[\text{bisulfate}]}$, meaning that the relative concentration of 1B is set to be 1. In the case of positively charged guanidine and dimethylamine clusters, the relative cluster concentrations are calculated based on protonated bases (1G1P and 1D1P) as $\frac{[\text{cluster}]}{[\text{protonated base}]}$, and for positively charged ammonia clusters, due to the absence of 1N1P in the mass spectrum, cluster ratio is calculated based on the smallest cluster detected as $\frac{[\text{cluster}]}{[\text{1A1N1P}]}$.

As discussed above, the interaction between bisulfate and sulfuric acid is very strong, and thus small anionic sulfuric acid clusters are very stable, having the largest relative concentrations in all cases. Guanidine is the only base with an interaction

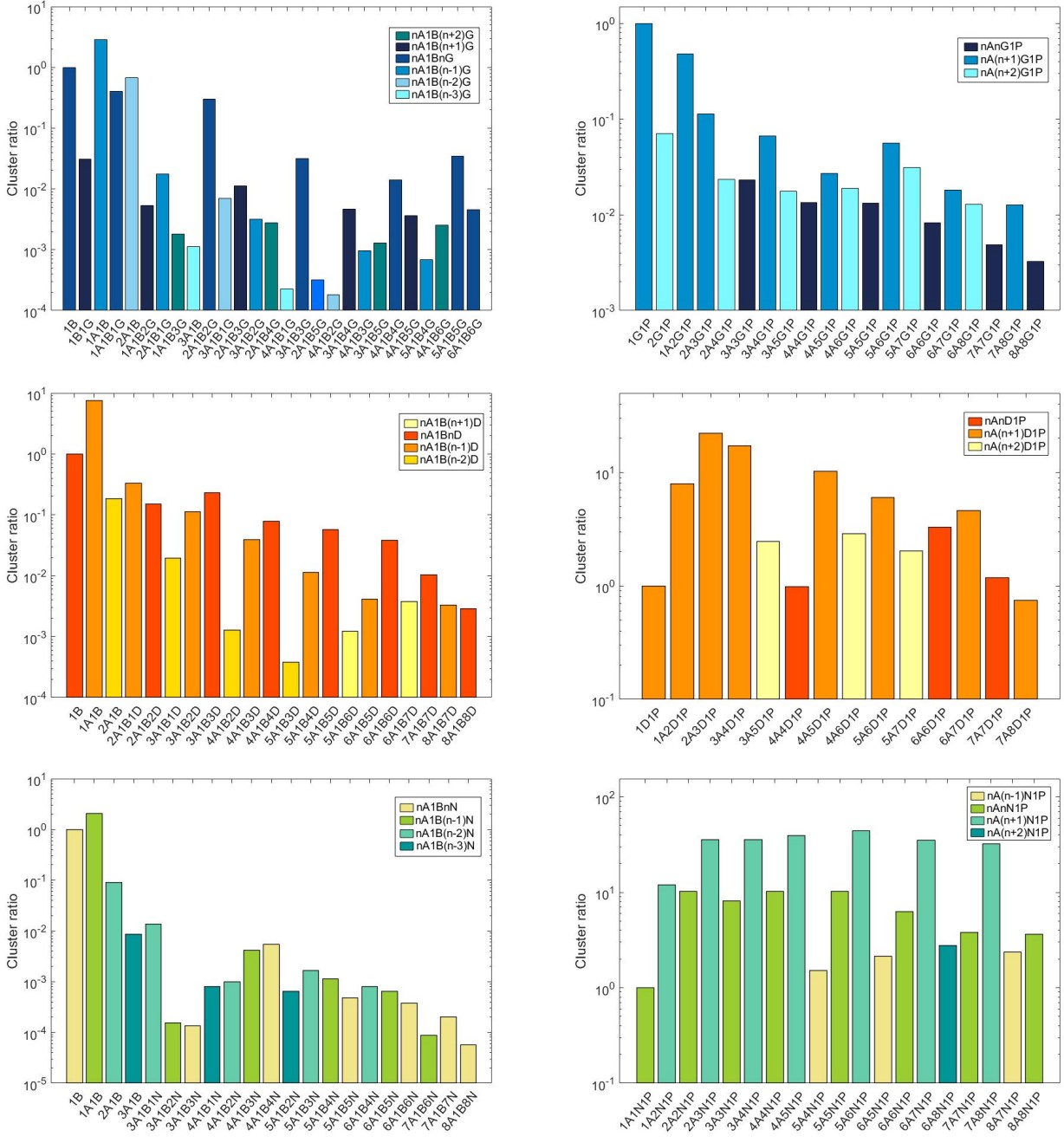

**Figure 8.** Experimentally determined relative cluster concentrations in negative (left) and positive (right) modes for sulfuric acid with guanidine (top), dimethylamine (middle) and ammonia (bottom).

strength with small anionic sulfuric acid clusters comparable to that of sulfuric acid, and thus 1B1G and 1B1A$n$G clusters can be detected. In the case of ammonia and dimethylamine, the smallest anionic base-containing clusters are 3A1B1N and 2A1B1D, respectively. The most abundant guanidine-containing clusters have the composition $n$A1B$n$G, and are detected up to $n$=6. Also clusters with a larger number of base compared to acid ($n$A1B($n$+1)G and $n$A1B($n$+2)G) can be detected. The negatively charged clusters consisting of dimethylamine and sulfuric acid are predicted to be less stable than the guanidine–sulfuric acid clusters (e.g. the evaporation rate for 4A1B4G is $2 \times 10^{-9}$ s$^{-1}$, and for 4A1B4D, it is eight orders of magnitude higher, $2 \times 10^{-1}$ s$^{-1}$). Accordingly, for dimethylamine, a smaller number of clusters is detected. The most abundant clusters are $n$A1B$n$D and $n$A1B($n$-1)D, similar to the trend observed for the computational results, and clusters up to $n$=8 are detected. Similarly to guanidine, $n$A1B($n$+1)D clusters are also observed, but only for $n \geq 5$. However, for ammonia, anionic clusters only with equal or smaller number of ammonia than sulfuric acid molecules are stable enough to be detected. These cover sizes from 3A1B1N to 8A1B8N.

In the negative mode, we always observe the formation of the sulfuric acid–bisulfate complex (1A1B). In the positive mode, the formation of 2(base)1P complex is also thermodynamically the most favorable first step in cluster formation and 2(base)1P clusters have low evaporation rates, but we observe the pure dimer cluster only for guanidine (2G1P). The absence of 1N1P and 2N1P in the mass spectrum is likely to be due to instrument limitation: the sensitivity of the instrument has been demonstrated to decrease dramatically for low mass-to-charge ratios (Heinritzi et al., 2016). The absence of 2D1P in the mass spectrum could be explained by the rapid formation of 1A2D1P, which is detected. As the theoretical data show, the addition of sulfuric acid to 2D1P is thermodynamically highly favorable ($-24.9$ kcal/mol), and the evaporation rate of 1A2D1P is seven orders of magnitude lower than that of 2D1P. In general, a larger number of clusters is observed in the negative than in the positive mode for all the analysed acid–base clusters. In the positive mode, most of the observed clusters contain more than 4 acids and 4 bases. In all cases, the diagonal clusters $n$(acid)($n$+1)(base)1P are the most abundant, which is in agreement with theoretical results. Also $n$A$n$N1P clusters are detected in the case of ammonia, whereas for dimethylamine and guanidine, the first detectable clusters below the diagonal axis are 4A4D1P and 3A3G1P, respectively. In addition, cationic clusters with two more neutral acid than base molecules ($n$A($n$-1)N1P) are detected only for ammonia. This could be explained by the fact that, in general, sulfuric acid forms more stable positively charged clusters with ammonia compared to dimethylamine and guanidine (e.g., the evaporation rates for 1A1N1P, 1A1D1P and 1A1G1P are $6 \times 10^{-1}$, $2 \times 10^3$ and $10^4$ s$^{-1}$, respectively).

### 3.5 Ion-Mediated Particle Formation

The roles of ammonia and dimethylamine in sulfuric-acid-driven NPF in the presence of ions have been studied in many laboratory experiments. Here we use our full cluster sets including both neutral and charged clusters, and compare the simulated NPF rates against those observed at the CLOUD chamber under similar conditions (Almeida et al., 2013; Kürten et al., 2018). Figure 9 shows experimental and theoretical particle formation rates at vapor concentrations of [D]=3–140 ppt$_V$ and [N]=10–15 ppt$_V$. In the simulations, the generic ion production rate is set to 3 cm$^{-3}$ s$^{-1}$ and the ion–wall loss enhancement factor to 3.3. (see Almeida et al. (2013)).

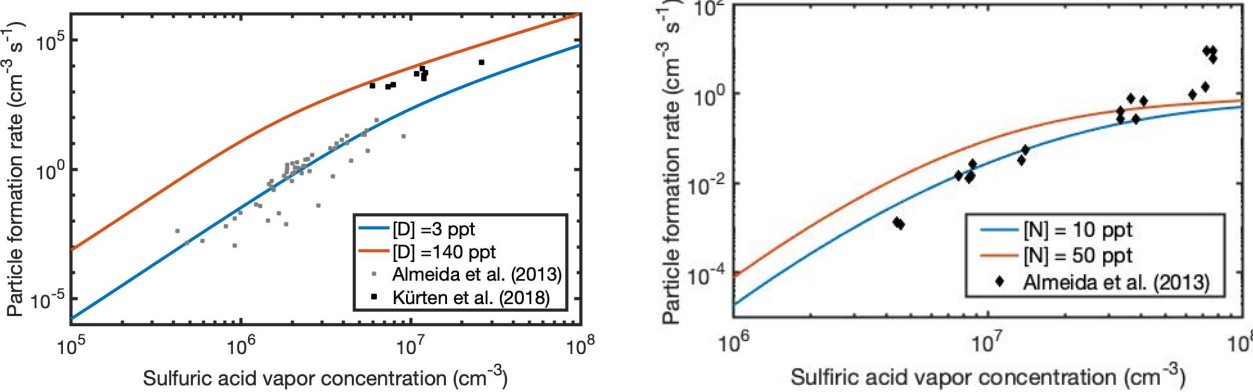

**Figure 9.** Particle formation rates observed at the CLOUD4 chamber experiment (markers) as a function of sulfuric acid vapor concentration at [D]=3–140 ppt$_V$ (left) and [N]=10–50 ppt$_V$ (right), and ACDC simulation results for particle formation in the presence of ions (lines). Note the different scales of $x$ and $y$ axes.

Figure 9 shows that the agreement between simulated and measured rates is good for both dimethylamine- and ammonia-containing clusters. In the case of dimethylamine, the measured data points from the study of Almeida et al. (2013) are close to the simulated [D]=3 ppt$_V$ particle formation rate. However, recently Kürten et al. (2018) reanalyzed that data using a more advanced method. The reanalysis yielded an order of magnitude higher particle formation rates than previously assessed, and
the reevaluated particle formation rates show a good agreement with our upper limit simulations [D]=140 ppt$_V$. For ammonia some experimental data points are ca. an order of magnitude higher than the simulated NPF rates, due to the plateauing of the simulated rates. This might be related to the effect of water in the experiments, as hydration is not considered in the present simulations. It has been shown that the effect of hydration is larger for clusters containing ammonia than for those containing dimethylamine. This is due to structural effects, such as the number of available hydrogen bond donors and acceptors within
the cluster (Yang et al., 2018).

There are no measured particle formation data for guanidine, however, the good agreement between the simulations and experiments for dimethylamine and ammonia indicates that also the simulations for guanidine can be considered realiable. Figure 10 shows the NPF rates for [G]=0.001–10 ppt$_V$ in the presence of ions with the same simulation conditions as for Figure 9, as well as a comparison to ion-free simulations for all the studied bases using the base concentration of 1 ppt$_V$.
For guanidine, the enhancing effect of ions on the NPF rate is very small. This is due the fact that the electrically neutral clusters are already so stable that further stabilization by ionic molecules does not have a significant effect. In the case of dimethylamine, the presence of ions increases the NPF rate by up to an order of magnitude at low acid concentrations (although at these conditions even the increased NPF rate is atmospherically very low). For ammonia, the effect of ions is crucial, leading to up to 20 orders of magnitude increase in the NPF rate. This is largely due to the generation of neutral cluster formation by
small ion cluster recombination, which allows clusters to "jump" over the unstable neutral cluster combinations. Since the thermodynamic barriers for anionic and cationic cluster growth are significantly lower compared to the neutral ammonia–sulfuric

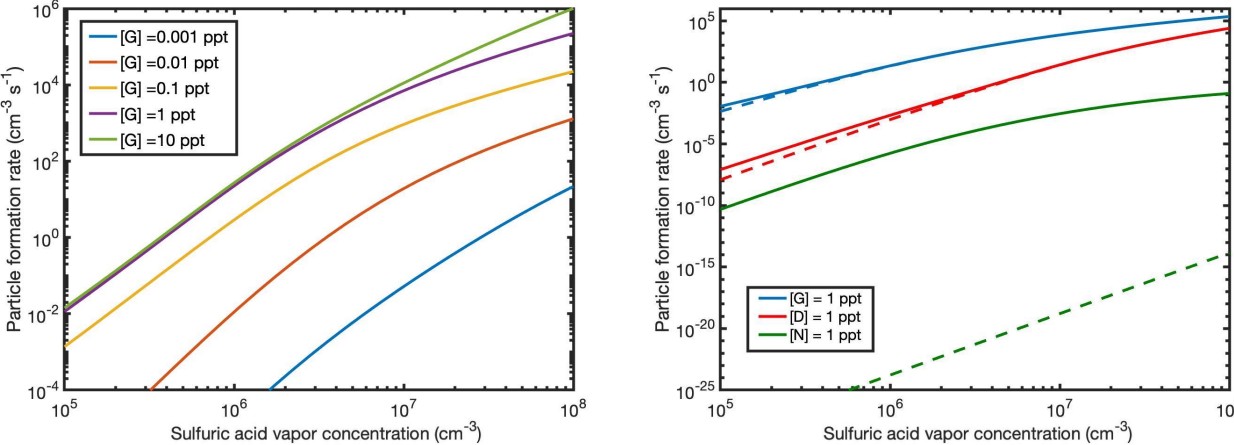

**Figure 10.** Particle formation rates from sulfuric acid and guanidine at [guanidine]=0.001–10 ppt$_V$ in the presence of ions (left), and ion-mediated (solid line) and neutral (dashed line) particle formation rates for guanidine, dimethylamine and ammonia at base concentrations of 1 ppt$_V$ (right) as a function of sulfuric acid vapor concentration.

acid case, larger clusters can form via collisions of medium-sized positively- and negatively-charged clusters. It should be noted that throughout the paper we have focused on *thermodynamic barriers*. In addition, the cluster growth might be hindered due to *kinetic barriers*. The addition of a monomer or a cluster to a pre-existing cluster might require cluster reorientation which in turn may lead to the breaking of intermolecular bonds, and thus, non-negligible kinetic barriers (DePalma et al., 2014; Bzdek et al., 2017; Xu et al., 2017). Thus, especially in the case of strongly-bound cage-like clusters, the subsequent growth as well as the evaporation may be slower than our calculations assume. For steady-state particle formation, however, the role of kinetic barriers is reduced assuming that the cluster formation and decomposition rate coefficients are connected by detailed balance. Because of this and for the fact that investigating all the possible barriers for formation and decomposition reactions is computationally very demanding, the kinetic barriers are neglected in this study.

## 3.6 Time-Dependent Simulations

It must be noted, however, that the total number of formed particles in a given time at atmospheric conditions is affected also by the time-dependent vapor concentrations. It is normally assumed that molecular clusters are a negligible sink of vapors, and that the concentration of vapor available for particle formation is determined by the vapor sources and the condensation sink onto larger particles. However, in the case of strongly clustering species and suppressed cluster evaporation, small clusters may take up a notable fraction of the vapor, leading to a negative feedback on particle formation due to vapor depletion. Therefore, in the atmosphere the particle formation efficiency of strong bases may be reduced compared to the steady-state predictions corresponding to constant vapor concentrations (Figures 10 and S2).

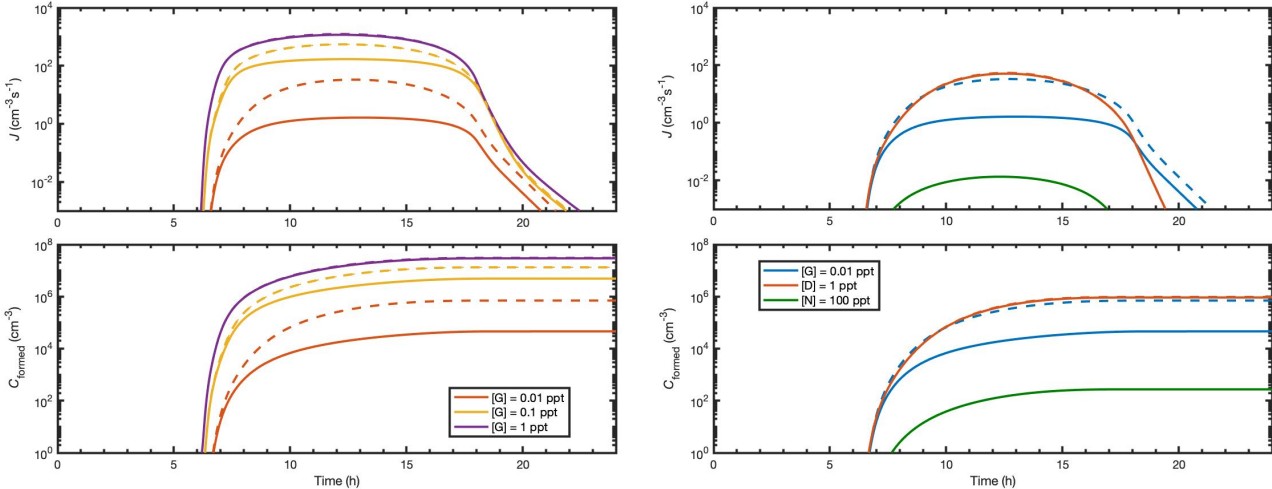

**Figure 11.** Particle formation rate $J$ and concentration $C_{\text{formed}}$ of formed particles as a function of time when either the base source is constant (solid lines) or the base concentration is constant (dashed lines).

Figure 11 demonstrates the effect of base vapor reduction due to clustering during a diurnal cycle where the source of sulfuric acid vapor is set to follow a sinusoidal function, mimicking atmospheric production of $H_2SO_4$ from $SO_2$ due to sunlight, and other parameters are set as in Figure S2. Dashed lines show the time-dependent particle formation rate (upper panels) and the total number of formed particles (lower panels) for simulations including both neutral and ionic clusters at a constant

base concentration. Solid lines show results for a constant base source $Q_{\text{base}} = L_{\text{base}} \times [\text{base}]$ that corresponds to the constant base concentrations assuming that they are determined solely by the source and the condensation sink $L_{\text{base}}$. For ammonia and dimethylamine, small clusters do not act as a notable sink to vapors, and the dashed and solid lines are indistinguishable from each other (right-hand side panels). For guanidine, however, a significant fraction of vapors can be bound to clusters, and neglecting this sink leads up to 1–2 orders of magnitude overprediction in the concentrations of formed particles (dashed vs.

solid lines in the lower panels). This effect is particularly significant at low vapor levels, but becomes negligible at a higher guanidine concentration or source (lines with different colors in the left-hand side panels).

While the effects depicted in Figure 11 do not affect steady-state particle formation investigations, such as the CLOUD set-up, they should be considered in the interpretation of field measurements and predictions of ambient aerosol formation, if there is reason to believe that strongly clustering species may be present and their sources can be assessed. Time-dependent clustering

simulations can be embedded in, for instance, an atmospheric box model to probe the vapor–cluster exchange dynamics. Such modeling approach can be applied to interpret field observations, and to estimate the vapor sink caused by clustering at different conditions.

## 4 Conclusions

We have investigated the enhancing potential of weak (ammonia), medium strong (dimethylamine) and very strong (guanidine) bases in atmospheric particle formation. In the studied sulfuric acid–base systems, molecular cluster growth proceeds through "diagonal" cluster compositions that contain approximately equal numbers of acid and base molecules. However, the difference between dimethylamine or ammonia and guanidine is that the growth of clusters containing the relatively weaker bases occurs via monomeric acid and base additions, whereas guanidine clusters mainly grow via acid–base heterodimer additions. This is because guanidine and sulfuric acid can form a complex with an evaporation rate as low as $10^{-5}$ s$^{-1}$, meaning that the probability for the complex to collide with an other heterodimer or a larger cluster is much higher than its probability to evaporate. We studied the structures of the diagonal clusters and showed that guanidine and sulfuric acid form extremely stable mesh-like cluster structures, which have a high symmetry and a large number of strong intermolecular interactions between bisulfate and guanidinium while the hydroxyl groups of the bisulfates remain free. Dimethylamine and ammonia form less symmetric structures with sulfuric acid, and the hydroxyl groups of the bisulfates form intermolecular interactions with each other. It is clear that the interaction between a bisulfate and a protonated base is much stronger than that between two bisulfates, and thus, the intermolecular interactions in the dimethylamine and ammonia clusters are much weaker than those in the guanidine clusters. By using relative base concentrations calculated from mass-balance relation, corresponding to the same equilibrium concentration for different heterodimers, we showed that the enhancing potential of a base is largely dominated by the intermolecular interactions between the acid and base molecules, and the atmospheric abundance of the base plays only a minor role in terms of cluster stability. Due to the fact that unprotonated guanidine is a semi-volatile compound, its actual atmospheric concentration may be much higher than the values used in this theoretical study. Also other strong base compounds, such as amidines and guanidine derivatives, are likely to have a higher enhancing potential in particle formation compared to medium strong bases such as alkylamines.

We compared simulated particle formation rates, based on the calculated Gibbs free energies, to rates measured in field and laboratory studies and found a good agreement for sulfuric acid–ammonia and sulfuric acid–dimethylamine particle formation. In addition, we compared simulated NPF rates with or without ions for the three representative bases, and demonstrated that in ion-mediated particle formation, the role of base strength is much smaller than in electrically neutral cases. In the case of ammonia, the enhancing effect of ions is significant, increasing the NPF rates by up to 20 orders of magnitude. For ammonia, the main neutral cluster growth pathway contains unstable clusters with evaporation rates up to $10^5$ s$^{-1}$, whereas the growth by formation of smaller anionic and cationic clusters and their subsequent recombination to larger neutral clusters occurs via stable clusters with much lower evaporation rates. In the case of guanidine, electrically neutral clusters are already tightly bound, and therefore, the effect of ions is very small.

Under atmospheric conditions, both ammonia and dimethylamine can be available, and new-particle formation may occur via three-component pathways. This three-component pathway can lead to higher new-particle formation rates than the two-component pathways for two main reasons: 1) base exchange and 2) synergistic effects. 1) For pre-existing sulfuric acid–ammonia clusters, the substitution of ammonia by dimethylamine can be assumed to be fast based on studies by e.g. Bzdek

et al. (2017) and Kupiainen et al. (2012). 2) When sulfuric acid–dimethylamine clusters uptake ammonia molecules, the number of intermolecular bonds increases which can further stabilize the clusters and thus make the subsequent cluster growth faster, as we have showed recently (Myllys et al., 2019). The role of base exchange and synergy are yet unresolved in the case of guanidine; however, since guanidine is a stronger base and capable of forming more intermolecular bonds than ammonia or dimethylamine, the reasonable assumption would be that guanidine can replace either ammonia or dimethylamine fast and that the replacement increases the cluster stability and particle formation rate. As sulfuric acid and guanidine form very stable clusters containing a large number of intermolecular bonds, we do not expect that the presence of either ammonia or dimethylamine would enhance the particle formation by synergistic effects.

Due to the substantial computational effort required, water is not included in the cluster structures or particle formation simulations of this study. It has been demonstrated that the enhancing effect of hydration is larger in the case of ammonia than dimethylamine (Olenius et al., 2017), and the reason is likely to be the number of available hydrogen bonds in the cluster structure (Yang et al., 2018). While the effect of water on guanidine–sulfuric acid particle formation remains to be resolved, the possible enhancement can be expected to be small as the unhydrated clusters are already extremely stable. Our study shows that the role of base strength and cluster structure, which affect the number and strength of intermolecular interactions, are often more important than differences in the typical atmospheric concentrations of different bases for steady-state particle formation. Therefore, when investigating the importance of acid–base chemistry on the formation and properties of atmospheric aerosol particles, impacts of strong organobases with very low concentrations should be included. The atmosphere is a complex mixture containing various potential contributors to NPF and identifying all the most relevant compounds and investigating their particle formation efficiency at different relative humidities is mandatory in order to understand and predict the importance of acid–base NPF in different atmospheric environments.

*Author contributions.* The study design, most of the calculations and simulations, and manuscript preparation were performed by NM. New configurational sampling were performed by JK and VB, experiments were performed and analyzed by DA and MP, time-dependent simulations were performed and analyzed by TO, and insightful discussion followed by valuable suggestions were given by JS. All co-authors proofread and commented the manuscript.

*Competing interests.* The authors declare that they have no conflict of interests.

*Acknowledgements.* We thank the Academy of Finland and ERC project 692891-DAMOCLES for funding, and the CSC-IT Center for Science in Espoo, Finland, for computational resources. NM thanks the Jenny and Antti Wihuri foundation for financial support. JS acknowledges funding from the U.S. National Science Foundation under grant no. CHE-1710580. TO thanks the ÅForsk foundation (project 18-334) and the Knut and Alice Wallenberg foundation (Academy Fellowship AtmoRemove) for funding. We thank professor R.B. Gerber for helpful discussion and doctor A. Kürten for valuable suggestion to use the reevaluated CLOUD data.

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
