# Peer review of "Role of Base Strength, Cluster Structure and Charge in Sulfuric Acid-Driven Particle Formation"

_Atmospheric Chemistry and Physics, 2019_

## Referee Comment (RC1) · Wei Huang (Referee) · 13 May 2019

This work is based on quantum chemical calculations and kinetic simulation as well as ESI-MS experiments to study the aerosol nucleation abilities for various composition and charge states. The cluster binding patterns with size dependence, the nucleation barrier analysis, the cluster relative concentrations ratio and the formation rates are properly analyzed. The strong binding of guanidine comparing with the other bases is highly emphasized and properly validated. This work provides important insights to the different base contributing nucleation abilities and has shown significant evidences from the view of structure, thermodynamics and kinetics. So I recommend it to be

published in Atmos. Chem. Phys. with minor revisions.

Specific comments:

1) In 3.3, for the conclusion that "This indicates that the enhancing effect of bisulfate in particle formation becomes more remarkable for weaker bases, since the presence of bisulfate removes the thermodynamic barrier of cluster growth (which does not exist in the case of guanidine)", it is unreasonable to draw this since the analysis for figure 6 only points to the stronger nucleation abilities with the increase of alkalinity for the case of bisulfate. If "the enhancing effect of bisulfate" could be known, it should be compared with different acids with the same base, like comparing figure 6 with figure 2. Whether or not the enhancing effect of bisulfate is more remarkable for weaker bases could also be figured out through comparing figure 6 with figure 2.

2) For the simulation of formation rates with the aid of ACDC, the boundary condition setting is crucial so the boundary conditions for all the systems should be added on the supplementary information.

3) The overestimation of new particle formation rates with constant base source is a very important conclusion. More descriptions about the constant base concentration case as well as the validity of simulation settings should be given.

Technological errors:

1) I am wondering if the free energies shown on figure 6 are standard Gibbs free energies or the actual concentration-dependent ones. If it's the former case, please correct it to match the figure caption.

---

## Referee Comment (RC2) · Anonymous Referee #2 · 14 May 2019

This paper reports structures and formation rates for new particles containing sulfuric acid and ammonia, dimethylamine, and guanidine, for neutral and charged clusters. The authors present a number of interesting findings, particularly that clusters can be a non-negligible sink of low-concentration but efficient stabilizing gases. The classification of growth mechanisms as a function of charge state and base is also a valuable contribution, and including these in one paper in one set of conditions helps to clarify the differences between these processes. I enthusiastically recommend publication and hope the authors will consider the following comments:

1. The paper mentions pKb but this is not the only measure of base strength. The

gas-phase proton affinities of the bases is certainly of relevance here, and makes it easier to separate effects from hydrogen bonding (solution-phase basicity) from effects purely due to site-specific basicity. In these clusters, the enhancing effects are likely a different combination of these two depending on the base, as guanidinium in particular can make significantly more hydrogen bonds than dimethylaminium. Can these results be broken down in this way?

2. What is the expected role of barriers between each growth step. Particularly for the heavily hydrogen-bonded cage structures formed in the guanidinium clusters, one might expect a substantial barrier (transition state) to addition of an additional monomer as a result of the need to break several strong hydrogen bonds to accommodate the new molecule. The paper mentions barriers several times, but this seems to be in reference to stable minima for given compositions that are uphill from a growth perspective. The presence of large barriers could certainly change the outcome for ammonia, and perhaps for the other two bases as well.

3. Over the range of concentrations studied at CLOUD, the simulations appear to overpredict the formation rates. Can the authors comment on this, particularly since the potential enhancing effects of water and mixed bases are not treated here?

4. The authors speculate on the role of water, but what about the role of base exchange?

Minutae: The term "intermolecular interactions" is used frequently in this paper, but it seems to be primarily referring to hydrogen bonds. Is there a specific reason the use of "hydrogen bond" is avoided here?

On page 3, a reference (gua, 2013) appears to be misformatted.
* * *

---

## Author Comment (AC1) · 21 Jun 2019

The comment was uploaded in the form of a supplement:
https://www.atmos-chem-phys-discuss.net/acp-2019-305/acp-2019-305-AC1-supplement.pdf

---

## Author Response (AR1)

BERKELEY • DAVIS • IRVINE • LOS ANGELES • MERCED • RIVERSIDE • SAN DIEGO • SAN FRANCISCO

SANTA BARBARA • SANTA CRUZ

**Department of Chemistry**

1102 Natural Sciences 2 University of California, Irvine Irvine, CA 92697-2025

**Dear Editor,**

we highly appreciate the constructive comments from the reviewers, and we have addressed the comments in the revised paper. We hope that the following responses are satisfying and that the paper can be accepted for publication in Atmos. Chem. Phys. The reviewers' comments have been reproduced in blue text below, followed by our point-by-point replies, and all changes to the manuscript text are also marked in blue in the revised version.

**Reviewer comments:**

**Reviewer: 1**

This work is based on quantum chemical calculations and kinetic simulation as well as ESI-MS experiments to study the aerosol nucleation abilities for various composition and charge states. The cluster binding patterns with size dependence, the nucleation barrier analysis, the cluster relative concentrations ratio and the formation rates are properly analyzed. The strong binding of guanidine comparing with the other bases is highly emphasized and properly validated. This work provides important insights to the different base contributing nucleation abilities and has shown significant evidences from the view of structure, thermodynamics and kinetics. So I recommend it to be published in Atmos. Chem. Phys. with minor revisions.

**Specific comments:**

1) In 3.3, for the conclusion that "This indicates that the enhancing effect of bisulfate in particle formation becomes more remarkable for weaker bases, since the presence of bisulfate removes the thermodynamic barrier of cluster growth (which does not exist in the case of guanidine)", it is unreasonable to draw this since the analysis for figure 6 only points to the stronger nucleation abilities with the increase of alkalinity for the case of bisulfate. If "the enhancing effect of bisulfate" could be known, it should be compared with different acids with the same base, like comparing figure 6 with figure 2. Whether or not the enhancing effect of bisulfate is more remarkable for weaker bases could also be figured out through comparing figure 6 with figure 2.

**Author reply:**

We agree that this sentence might be confusing. We have moved the sentence to the end of 3.3 and modified it to the following format:

"The enhancing effect of ions in particle formation becomes more remarkable for weaker bases, since the presence of ions allows the cluster growth to occur without thermodynamic barrier (which does not exist in the case of guanidine). The enhancing effect of ions on particle formation rates is presented in Figure 10."

2) For the simulation of formation rates with the aid of ACDC, the boundary condition setting is crucial so the boundary conditions for all the systems should be added on the supplementary information.

**Author reply:**

We thank for the suggestion. We have added the boundary conditions to the supporting information.

**3) The overestimation of new particle formation rates with constant base source is a very important conclusion. More descriptions about the constant base concentration case as well as the validity of simulation settings should be given.**

**Author reply:**

Modeling of particle formation rates is generally based on the assumptions that the rate is in a steady state, and that the small clusters do not reduce or deplete the vapors. However, these assumptions may not always be valid for real atmospheric situations, which is demonstrated in Figure 11 in terms of the base concentration. Since chamber experiments, such as CLOUD, address steady-state situations, these effects do not concern formation rates measured and modeled for such set-ups. Instead, the two main implications apply to 1) interpretation of field observations, and 2) atmospheric model predictions of vapor concentrations and particle numbers originating from gas-to-particle conversion. First, if observed atmospheric formation events are driven by very strongly clustering species, comparisons with model predictions need to involve explicitly time-dependent cluster formation simulations. Ideally, these include estimates of the vapor sources, if available (see e.g. Hemmilä et al. (2018)). Second, clustering dynamics simulations can also be embedded in an atmospheric model framework for predictions of time-dependent vapor and nanoparticle concentrations given that the computational demand is not too high. These results can also be used to derive approximations for computationally heavier applications, such as large-scale chemical transport and climate models. This is now discussed in Section 3.6.

"While the effects depicted in Figure 11 do not affect steady-state particle formation investigations, such as the CLOUD set-up, they should be considered in the interpretation of field measurements and predictions of ambient aerosol formation, if there is reason to believe that strongly clustering species may be present and their sources can be assessed. Time-dependent clustering simulations can be embedded in, for instance, an atmospheric box model to probe the vapor–cluster exchange dynamics. Such modeling approach can be applied to interpret field observations, and to estimate the vapor sink caused by clustering at different conditions."

**Technological errors:**

1) I am wondering if the free energies shown on figure 6 are standard Gibbs free energies or the actual concentration-dependent ones. If it's the former case, please correct it to match the figure caption.

**Author reply:**

Figure 6 presents the concentration-dependent Gibbs free energies at a sulfuric acid concentration of 107 cm-3 and the relative base concentrations. The corresponding Gibbs free binding energies, from which the actual Gibbs free energies are calculated, are presented in the supporting information Table S1.

**Reviewer: 2**

This paper reports structures and formation rates for new particles containing sulfuric acid and ammonia, dimethylamine, and guanidine, for neutral and charged clusters. The authors present a number of interesting findings, particularly that clusters can be a non-negligible sink of low-concentration but efficient stabilizing gases. The classification of growth mechanisms as a function of charge state and base is also a valuable contribution, and including these in one paper in one set of conditions helps to clarify the differences between these processes. I enthusiastically recommend publication and hope the authors will consider the following comments:

1) The paper mentions pKb but this is not the only measure of base strength. The gas-phase proton affinities of the bases is certainly of relevance here, and makes it easier to separate effects from hydrogen bonding (solution-phase basicity) from effects purely due to site-specific basicity. In these clusters, the enhancing effects are likely a different combination of these two depending on the base, as guanidinium in particular can make significantly more hydrogen bonds than dimethylaminium. Can these results be broken down in this way?

**Author reply:**

This question covers a very important shortcoming in atmospheric clustering studies. We hope to fully respond to this question in a separate publication in the near future. For now, we have added the gas-phase basicities to the Introduction to give an additional measure for base strength.

2) What is the expected role of barriers between each growth step. Particularly for the heavily hydrogenbonded cage structures formed in the guanidinium clusters, one might expect a substantial barrier (transition state) to addition of an additional monomer as a result of the need to break several strong hydrogen bonds to accommodate the new molecule. The paper mentions barriers several times, but this seems to be in reference to stable minima for given compositions that are uphill from a growth perspective. The presence of large barriers could certainly change the outcome for ammonia, and perhaps for the other two bases as well.

**Author reply:**

Indeed, this paper does not address the possible kinetic barriers between global minimum cluster structures. We agree that such kinetic barriers may be non-negligible, both in cluster formation and decomposition. Thus, especially in the case of strongly-bound cage-like clusters, the subsequent growth as well as the evaporation are slower than our calculations assume. Since investigating all the possible barriers for formation and decomposition reactions is computationally very demanding, the possible activation energy barriers are neglected here as well as in most atmospheric clustering studies. Based on comparisons between experimental and theoretical results, however, the lack of activation barriers in simulations either does not affect remarkably the particle formation rates or is compensated by some other approximation leading to a beneficial cancellation of errors. Specifically, for steady-state particle formation, the role of kinetic barriers is reduced assuming that the cluster formation and decomposition-dependent. Therefore, in this study we do not speculate on the role of activation energy barriers, but some useful guide can be found e.g. in the studies by DePalma et al. (2014), Xu et al. (2017) and Bzdek et al. (2017). We have added the following in the end of Section 3.5:

"It should be noted that throughout the paper we have focused on *thermodynamic barriers*. In addition, the cluster growth might be hindered due to *kinetic barriers*. The addition of a monomer or a cluster to a preexisting cluster might require cluster reorientation which in turn may lead to the breaking of intermolecular bonds, and thus, non-negligible kinetic barriers (DePalma et al., 2014; Bzdek et al., 2017; Xu et al., 2017). Thus, especially in the case of strongly-bound cage-like clusters, the subsequent growth as well as the evaporation may be slower than our calculations assume. For steady-state particle formation, however, the role of kinetic barriers is reduced assuming that the cluster formation and decomposition rate coefficients are connected by detailed balance. Because of this and for the fact that investigating all the possible barriers for formation and decomposition reactions is computationally very demanding, the kinetic barriers are neglected in this study."

3) Over the range of concentrations studied at CLOUD, the simulations appear to overpredict the formation rates. Can the authors comment on this, particularly since the potential enhancing effects of water and mixed bases are not treated here?

**Author reply:**

The reason for this may actually be an underprediction of the experimental CLOUD particle formation rates by the analysis of Almeida et al. (2013). Kürten et al. (2018) reanalyzed the sulfuric acid-dimethylamine particle formation rates for the data measured earlier at CLOUD using a more advanced method. The reanalysis yielded an order of magnitude higher particle formation rates than previously assessed, and we have now

added these new CLOUD data to Figure 9. The reevaluated particle formation rates show a good agreement with our upper limit simulations ([dimethylamine]=140  $ppt_V$ ). We have updated the figure and added the following:

"In the case of dimethylamine, the measured data points from the study of Almeida et al. (2013) are close to the simulated  $[D]=3 \text{ ppt}_{V}$  particle formation rate. However, recently Kürten et al. (2018) reanalyzed that data using a more advanced method. The reanalysis yielded an order of magnitude higher particle formation rates than previously assessed, and the reevaluated particle formation rates show a good agreement with our upper limit simulations  $[D]=140 \text{ ppt}_{V}$ .

4) The authors speculate on the role of water, but what about the role of base exchange?

**Author reply:**

We agree that the role of base exchange as well as base synergy is worth of mention and we have added the following paragraph in the Conclusions section:

"Under atmospheric conditions, both ammonia and dimethylamine can be available, and new-particle formation may occur via three-component pathways. This three-component pathway can lead to higher new-particle formation rates than the two-component pathways for two main reasons: 1) base exchange and 2) synergistic effects. 1) For pre-existing sulfuric acid-ammonia clusters, the substitution of ammonia by dimethylamine can be assumed to be fast based on studies by e.g. Bzdek et al. (2017) and Kupiainen et al. (2012). 2) When sulfuric acid-dimethylamine clusters uptake ammonia molecules, the number of intermolecular bonds increases which can further stabilize the clusters and thus make the subsequent cluster growth faster, as we have showed recently (Myllys et al., 2019). The role of base exchange and synergy are yet unresolved in the case of guanidine; however, since guanidine is a stronger base and capable of forming more intermolecular bonds than ammonia or dimethylamine, the reasonable assumption would be that guanidine can replace either ammonia or dimethylamine fast and that the replacement increases the cluster stability and particle formation rate. As sulfuric acid and guanidine form very stable clusters containing a large number of intermolecular bonds, we do not expect that the presence of either ammonia or dimethylamine would enhance the particle formation by synergistic effects."

Minutae: The term "intermolecular interactions" is used frequently in this paper, but it seems to be primarily referring to hydrogen bonds. Is there a specific reason the use of "hydrogen bond" is avoided here?

**Author reply:**

We have used "intermolecular interactions" since it covers both: the hydrogen bond between neutral electronegative and -positive atoms as well as the ionic bond where the proton transfer has occurred.

On page 3, a reference (gua, 2013) appears to be misformatted.

**Author reply:**

We thank the reviewer for spotting this mistake, and we have corrected the reference.

Almeida, J., et al.: Molecular Understanding of Sulphuric Acid–Amine Particle Nucleation in the Atmosphere, Nature, 502, 359–363, 2013.

Bzdek, B., et al.: Mechanisms of Atmospherically Relevant Cluster Growth, Acc. Chem. Res., 50, 1965–1975, 2017.

DePalma, J., et al.: Activation Barriers in the Growth of Molecular Clusters Derived from Sulfuric Acid and Ammonia, J. Phys. Chem. A, 118, 11547–11554, 2014.

Kupiainen, O., et al.: Amine Substitution into Sulfuric Acid–Ammonia Clusters, Atmos. Chem. Phys., 12, 3591–3599, 2012.

Kürten, A., et al.: New Particle Formation in the Sulfuric Acid–Dimethylamine–Water System: Reevaluation of CLOUD Chamber Measurements and Comparison to an Aerosol Nucleation and Growth Model, Atmos. Chem. Phys., 18, 845–863, 2018.

Hemmilä, M., et al.: Amines in boreal forest air at SMEAR II station in Finland, Atmos. Chem. Phys., 18, 6367–6380, 2018.

Myllys, N., et al.: Molecular-Level Understanding of Synergistic Effects in Sulfuric Acid–Amine–Ammonia Mixed Clusters, J. Phys. Chem. A, 123, 2420–2425, 2019.

Xu, J., et al.: Nanoparticles Grown from Methanesulfonic Acid and Methylamine: Microscopic Structures and Formation Mechanism., Phys. Chem. Chem. Phys., **19**, 31949–31957, 2017.

**Role of Base Strength, Cluster Structure and Charge in Sulfuric Acid-Driven Particle Formation**

Nanna Myllys1,2, Jakub Kubečka2, Vitus Besel2, Dina Alfaouri2, Tinja Olenius3, James N Smith1, and Monica Passananti2,4

1Department of Chemistry, University of California, Irvine

[revised manuscript text omitted]

15
$$[(acid)(base)] = [acid][base] \frac{k_{\rm B}T}{P_{\rm ref}} \exp\left(-\frac{\Delta G_{\rm ref}}{k_{\rm B}T}\right)$$
 (1)

The equilibrium concentration [(acid)(base)] of the heterodimer is dependent both on the Gibbs free formation energy  $\Delta G_{ref}$  (calculated at reference pressure  $P_{ref}$ ) at given temperature T and on the monomer concentrations [acid] and [base]. Now we can study how large the magnitude of the exponential Gibbs free energy contribution has relative to the linear concentration factors. The Gibbs free formation energies (at 298.15 K) are -6.8 kcal/mol for ammonia–sulfuric acid, -13.5 kcal/mol for